# Human papillomavirus seroprevalence in pregnant women following gender-neutral and girls-only vaccination programs in Finland: A cross-sectional cohort analysis following a cluster randomized trial

Penelope Gray[1]*, Hanna Kann[2], Ville N. Pimenoff[2,3,4], Tiina Eriksson[5], Tapio Luostarinen[6], Simopekka Vänskä[7], Heljä-Marja Surcel[8,9], Helena Faust[2], Joakim Dillner[2], Matti Lehtinen[2,4,7,10]

1 Faculty of Social Sciences, Tampere University, Tampere, Finland, 2 Department of Laboratory Medicine, Karolinska Institutet, Stockholm, Sweden, 3 Oncology Data Analytics Program, Bellvitge Biomedical Research Institute (IDIBELL), Consortium for Biomedical Research in Epidemiology and Public Health (CIBERESP), Hospitalet de Llobregat, Barcelona, Spain, 4 FICAN Mid, Tampere, Finland, 5 Department of Research and Development, Tampere University Hospital, Tampere, Finland, 6 Finnish Cancer Registry, Helsinki, Finland, 7 Department of Infectious Disease Control and Vaccination, Finnish Institute for Health and Welfare, Helsinki, Finland, 8 Faculty of Medicine, University of Oulu, Oulu, Finland, 9 Biobank Borealis of Northern Finland, Oulu University Hospital, Oulu, Finland, 10 Infections and Cancer Epidemiology, German Cancer Research Center (DKFZ), Heidelberg, Germany

* penelope.gray@tuni.fi

## Abstract

### Background

Cervical cancer elimination through human papillomavirus (HPV) vaccination programs requires the attainment of herd effect. Due to its uniquely high basic reproduction number, the vaccination coverage required to achieve herd effect against HPV type 16 exceeds what is attainable in most populations. We have compared how gender-neutral and girls-only vaccination strategies create herd effect against HPV16 under moderate vaccination coverage achieved in a population-based, community-randomized trial.

### Methods and findings

In 2007–2010, the 1992–1995 birth cohorts of 33 Finnish communities were randomized to receive gender-neutral HPV vaccination (Arm A), girls-only HPV vaccination (Arm B), or no HPV vaccination (Arm C) (11 communities per trial arm). HPV16/18/31/33/35/45 seroprevalence differences between the pre-vaccination era (2005–2010) and post-vaccination era (2011–2016) were compared between all 8,022 unvaccinated women <23 years old and resident in the 33 communities during 2005–2016 (2,657, 2,691, and 2,674 in Arms A, B, and C, respectively). Post- versus pre-vaccination-era HPV seroprevalence ratios (PRs) were compared by arm. Possible outcome misclassification was quantified via probabilistic bias analysis. An HPV16 and HPV18 seroprevalence reduction was observed post-

**Data Availability Statement:** All the pertinent summary-level data are contained within the

manuscript and supplementary files. All other relevant underlying individual-level data will be returned to Northern Finland Biobank Borealis in accordance with the signed Material Transfer Agreement. Biobank Borealis will subsequently make this individual-level data available researchers in accordance with their data access policies (contact via: biopankkiborealis@ppshp.fi).

**Funding:** The study was also supported by grants from the Swedish Cancer Society (CAN 2015/399 and CAN 2017/459, https://www.cancerfonden.se/), from the Swedish Foundation for Strategic Research (grant number RB13-0011, https://strategiska.se/en/) and from Karolinska Institutet (Dnr. 2019-01523, https://ki.se/). ML received funding from KI for his Professorship (Dnr. 2-3698/2017, https://ki.se/). PG received personal working grants from the Cancer Society of Finland (pink ribbon fund, https://www.cancersociety.fi/) and the City of Tampere Science Fund (https://www.tampere.fi/). GlaxoSmithKline Biologicals SA funded the community-randomized HPV-040 trial (NCT00534638, https://www.gsk.com/) however was not in any way involved in the conduct of this study. The authors are solely responsible for the final content and interpretation.

**Competing interests:** I have read the journal's policy and the authors of this manuscript have the following competing interests: M.L. has previously received grants from Merck & Co. and GSK Biologicals through his employers the Finnish Institute of Health and Welfare (THL) and the University of Tampere for HPV vaccination studies.

**Abbreviations:** FMC, Finnish Maternity Cohort; HBV, hepatitis B virus; HPV, human papillomavirus; HSV-2, herpes simplex virus type 2; PR, seroprevalence ratio; RPR, ratio of seroprevalence ratios.

vaccination in the gender-neutral vaccination arm in the entire study population ($PR_{16}$ = 0.64, 95% CI 0.10–0.85; $PR_{18}$ = 0.72, 95% CI 0.22–0.96) and for HPV16 also in the herpes simplex virus type 2 seropositive core group ($PR_{16}$ = 0.64, 95% CI 0.50–0.81). Observed reductions in HPV31/33/35/45 seroprevalence ($PR_{31/33/35/45}$ = 0.88, 95% CI 0.81–0.97) were replicated in Arm C ($PR_{31/33/35/45}$ = 0.79, 95% CI 0.69–0.90).

## Conclusions

In this study we only observed herd effect against HPV16/18 after gender-neutral vaccination with moderate vaccination coverage. With only moderate vaccination coverage, a gender-neutral vaccination strategy can facilitate the control of even HPV16. Our findings may have limited transportability to other vaccination coverage levels.

## Trial registration

ClinicalTrials.gov number NCT00534638, https://clinicaltrials.gov/ct2/show/NCT00534638.

## Author summary

### Why was this study done?

- High-risk human papillomavirus (HPV) infection is a necessary cause of cervical cancer in females.

- HPV vaccination targeting high-risk HPV types 16 and 18 has been implemented internationally.

- Achieving herd protection for HPV16 may require greater than 80% girls-only vaccination coverage, a level that has not been achievable in many countries.

- We evaluate whether gender-neutral or girls-only HPV vaccination results in HPV16 and HPV18 herd protection when the vaccination coverage is only moderate (40%–50%).

### What did the researchers do and find?

- We implemented a community-randomized trial of gender-neutral versus girls-only versus no HPV vaccination of young adolescents in 2007–2010, with 11 communities in each arm. Vaccination coverage was implemented with moderate coverage (40%–50%) at the community level.

- We evaluated the herd effect created by the different vaccination strategies by measuring the cumulative incidence of vaccine-protected HPV types in 8,022 young unvaccinated pregnant females (under 23 years old), comparing the time periods 2005–2010 (pre-vaccination) and 2011–2016 (post-vaccination).

- An HPV16 herd effect, that is, a reduction in cumulative incidence among the unvaccinated females, was only observed in communities where gender-neutral vaccination had been implemented.

### What do these findings mean?

- Achieving a vaccination coverage of above 80%, which is required to achieve herd effect against HPV16, may be unrealistic in some populations. Implementing gender-neutral HPV vaccination provides a solution to this problem as the vaccination coverage threshold required to provide herd effect to unvaccinated females is lower.

- Our study finds that gender-neutral vaccination provides stronger herd effect than girls-only vaccination in the setting of moderate vaccination coverage. However, these findings are limited to this setting and are not readily generalizable to settings with high (>80%) vaccination coverage.

## Introduction

The World Health Organization has called for the elimination of cervical cancer as a public health problem [1]. To this end, the WHO has developed a global strategy requiring every country globally to achieve 90% human papillomavirus (HPV) vaccination of girls by the age of 15 years by the year 2030 [2]. However, although some countries such as Scotland have achieved 90% coverage, achieving this globally may be a near impossible challenge [3,4]. Present vaccination coverage levels [5] are notably below the 80% vaccination coverage that is required for the eradication of vaccine-targeted HPV types [6], and herd effect among unvaccinated individuals is needed.

   HPV vaccines provide not only strong direct protection but also herd effect/herd protection, also known as herd immunity (i.e., indirect protection to unvaccinated individuals) due to assortative transmission of the HPVs [7–10]. Modeling studies have suggested that already low to moderate vaccination coverage inclusion of boys provides incremental herd effect to unvaccinated girls [6,9,11,12]. In our unique community-randomized HPV16/18 vaccination trial, the herd effect/herd immunity created has been measured as the degree of decrease in HPV incidence/prevalence in unvaccinated women [7,13,14]. We found the predicted herd effect against vaccine-targeted HPV18, and cross-protection against HPV types 31, 33, 35, and 45, when vaccination coverage was approximately 50% [7,13,14]. In populations implementing girls-only vaccination, notable herd effect against HPV16 (the most oncogenic and most common HPV type) has only been observed when the vaccination coverage was high [15–17]. This is probably due to the high basic reproduction number (R0) of HPV16 compared to other HPV types [17], and may depend also on the method of identifying HPV occurrence (one-time PCR positivity or seropositivity, i.e., prevalence or cumulative incidence).

   We performed population-based HPV analysis to evaluate the herd effect created by gender-neutral or girls-only vaccination following our community-randomized trial in the instance of low to moderate vaccination coverage. In the previous reports of this trial, the herd effect was evaluated using transitory HPV PCR positivity in study participants when they were aged 18 and/or 22 years; although notable HPV18 herd effect was observed, no HPV16 herd effect was found [12,13,18]. To provide assurance that the lack of HPV16 herd effect was not due to the methodological approach, we then nested a cross-sectional cohort within the community-randomized trial. We then estimated HPV16 seroprevalence (cumulative HPV16 incidence) over time using pre-and post-vaccination-era sera from unvaccinated women under the age of 23 years and resident in the communities with gender-neutral or girls-only vaccination strategies. Possible clearance of an ecological niche by HPV16/18 vaccination is also now

described as the natural counterpart to the serology-based type-replacement study concerning non-vaccine HPV types [18].

## Methods

### Study design

A population-based, community-randomized HPV vaccination trial was conducted among female and male 1992–1995 birth cohorts between 2007 and 2010 [19]. The trial was originally designed to guide evidence-based decision-making regarding national HPV vaccination policy [20,21], by testing the primary hypothesis of difference in the creation of herd effect by gender-neutral versus girls-only HPV vaccination strategies. Thirty-three geographically distinct Finnish communities located a minimum of 50 km from the next nearest community (or 35 km in the case of the 5 communities from the Helsinki metropolitan area) were included in the trial. To increase study power, the coefficient of variation, Ks (Ks = 0.13), between communities was minimized by first stratifying the communities by previously ascertained HPV16/18 seroprevalence [22] into those with low, moderate, and high seroprevalence. From these 3 strata, the communities were then randomized using a random number generator to 3 trial arms: In Arm A communities, 90% of girls and boys received HPV vaccination, and 10% of girls and boys received hepatitis B virus (HBV) vaccination; in Arm B communities, 90% of girls received HPV vaccination, and 10% of girls and all boys received HBV vaccination; in Arm C communities, all girls and boys received HBV vaccination.

In total, 80,272 Finnish- or Swedish-speaking girls and boys in the 1992–1995 birth cohorts were identified via the Finnish Population Register Centre as being resident in the 33 trial communities. Out of this group, 20,513 girls and 11,662 boys participated in the trial with parental/guardian informed consent. The study was partially blinded to all Arm A study participants and all female Arm B participants. Vaccination took place from 2007 to 2010, when the participants were aged 12–15 years, with 99.4% of participants receiving all 3 doses of the allocated vaccine (the bivalent HPV vaccine Cervarix or the HBV vaccine Engerix-B). The mean community-level vaccination coverage acquired via this vaccination was 47.1% in Arm A communities and 45.8% in Arm B communities among girls from the 1992–1995 birth cohorts (standard deviation [SD] = 9.4% and 6.6%, respectively). In Arm A communities the vaccination coverage acquired among boys from the 1992–1995 birth cohorts was 19.5% (SD = 7.1%) [19,20].

The creation of herd effect by different HPV16/18 vaccination strategies over time was estimated via a nested cross-sectional cohort study [23] of all pregnant women under the age of 23 years who were resident in the 33 trial communities from 2005 until the end of 2016. Their serum samples were extracted from a population-representative biobank, the Finnish Maternity Cohort (FMC) [18,21]. The FMC biobank houses 2 million serum samples obtained from approximately all 1 million pregnant Finnish women between 1983 and 2016 for screening of congenital infections. The participating women provided informed consent at the maternity clinic to have their samples stored for research purposes by the FMC biobank; 96% of women consented.

FMC participants eligible for this study were under the age of 23 years at the time of sample donation, first-time donors to the FMC, resident in 1 of the 33 trial communities, and HPV unvaccinated [18]. In Finland, every citizen (or person resident for greater than 3 months) is given a unique personal identification number at birth (or shortly after arrival into the country). HPV vaccination status was confirmed by linkage via the participants personal identification number with the national HPV vaccination trial registry both prior to and after sample extraction. For the birth cohorts eligible to receive HPV vaccination via the Finnish national

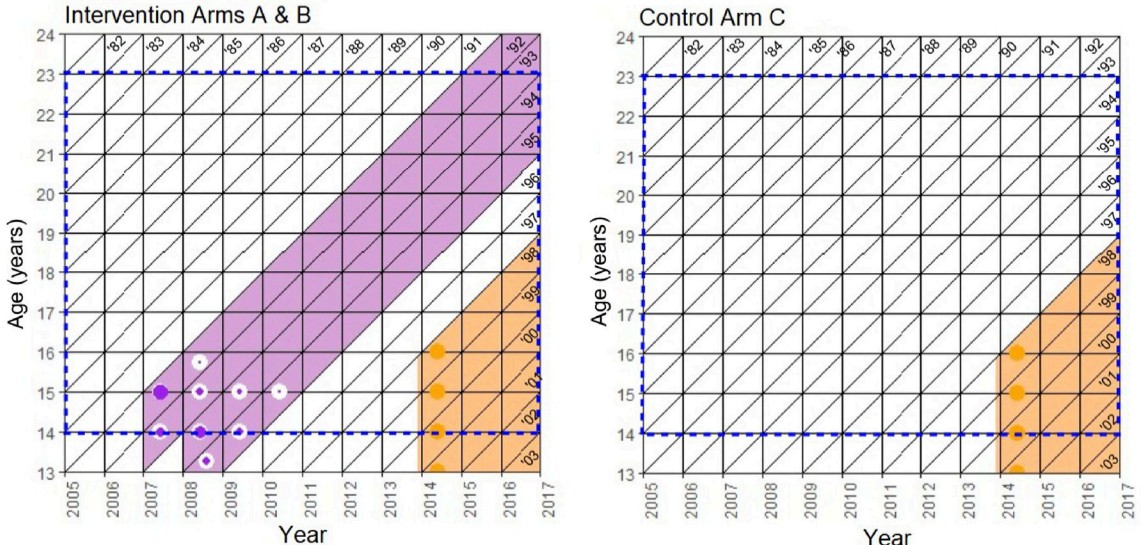

**Fig 1. Lexis diagrams depicting the community-level exposure of the adolescent population to direct and indirect effects of the cluster-randomized human papillomavirus vaccination trial by birth cohort and study arm.** hite bars represent the birth cohorts with no vaccination, and the purple (trial vaccination) and orange (national vaccination) bars represent post-vaccination birth cohorts. The blue dashed lines indicate the sampling years and ages of this study. The colored fill of the symbols indicates the proportion of each type of vaccination that took place at that time point and age per birth cohort.

vaccination program (1998 and younger birth cohorts), HPV vaccination status was ascertained by manually scrutinizing participants' HPV antibody levels for titers indicative of HPV vaccination (i.e., multiple-fold those acquired via natural infection for HPV16 and HPV18).

The eligible participants for the serosurvey came from the 1982 and younger birth cohorts. The 1992 to 1995 birth cohorts were exposed to community-level vaccination via the community-randomized trial intervention, and the 1998 and younger birth cohorts were exposed to community-level HPV vaccination via the Finnish national HPV vaccination program initiated in late 2013 (Fig 1 and S1 Fig). The sampling time frame was divided into the pre-vaccination period (2005–2010) and the post-vaccination period (2011–2016). All the pregnant females under the age of 23 years at the time of sample donation from each of the trial communities were included, totaling 8,022 females.

Data regarding self-reported maternal smoking among women under the age of 23 years and resident in the 33 communities between 2005 and 2016 were collected from the Finnish Medical Birth Register, and used as a surrogate of community-level risk-taking behaviors. To define the core group with high contact rate, we identified herpes simplex virus type 2 (HSV-2) seropositive women [18]. Data on community-specific vaccination coverage over each calendar year were collected from the HPV trial registry for the birth cohorts exposed to the community-randomized trial and from the Finnish vaccination register for the birth cohorts exposed to the Finnish national HPV vaccination program.

This study is reported in accordance to the Strengthening the Reporting of Observational Studies in Epidemiology (STROBE) guideline (S1 Checklist).

## Ethics

The community-randomized HPV-040 study obtained permissions from the Ethical Review Board of Pirkanmaa Hospital District (R07113M 14.6.2007). The FMC steering committee

granted permission for the linkage and use of the serum samples. No harm was caused to the cohorts.

## Laboratory analyses

The serum samples were analyzed for the presence of IgG antibodies to HPV types 6, 11, 16, 18, 31, 33, 35, 39, 45, 51, 52, 56, 58, 59, 66, 68, and 73 and HSV-2 using multiplexed heparin-bound pseudovirion (and HSV-2 glycoprotein gG2) Luminex assay [24]. Seropositivity cutoff levels were established with a negative control panel of serum samples from 191 children $\leq$12 years old (mean age = 4.7 years) (S1 Text).

## Statistical analyses

The primary hypothesis of this study was that HPV16/18 vaccination created a herd effect (in the HPV-040 and type-replacement study protocols, this was called "indirect effect" or "ecological niche formation") over time. In this study, herd effect is defined and measured as the degree of decrease in HPV cumulative incidence (unattributable to random or systematic error) among unvaccinated individuals in the post-vaccination era. To investigate the herd effect (indirect effect) of increasing community-level HPV16/18 vaccination during the study period (via a gender-neutral or girls-only vaccination strategy), we calculated the absolute seroprevalence of vaccine-targeted HPV types 16, 18, and 16/18 (combined), and vaccine-cross-protected HPV types 31, 33, 35, 45, and 31/33/35/45 (combined). This was calculated for the pre- and post-vaccination eras, 2005–2010 and 2011–2016, respectively. In the case of the former, the participants who had donated the sera were likely to have been unexposed to the indirect effects of HPV vaccination, whereas in the case of the latter, the participants may have been under herd effect [14,19,20].

The degree of clustering of HPV16/18 and HPV16/18/31/33/35/45 seropositivity was assessed by calculation of the intracluster correlation coefficient from the pre-vaccination-era data (from 2005 to 2010) using Fleiss and Cuzick's estimator in combination with Zou and Donner's modified Wald test to compute the 95% confidence intervals [25,26].

The exposure in this study is defined as exposure to the herd effects (indirect effects) of HPV16/18 vaccination due to residing at the time of sample donation in one of the communities of the community-randomized trial. Thus, to evaluate the extent of exposure in the study population of pregnant females under the age of 23 years, the birth-cohort-, community-, and year-specific vaccination coverage was calculated. From this, the community-specific vaccination coverage by year among the study population was then calculated as the birth-cohort-weighted vaccination coverage by gender, weighted by the proportion of participants from each birth cohort found in each year of the study among the study population of pregnant females. We also calculated HSV-2 seroprevalence to assess changes in the occurrence of sexually transmitted infections between the pre- and post-vaccination eras. Calendar-time-specific absolute seroprevalence was calculated stratified by Arms A, B, and C of the community-randomized trial. The accompanying 95% confidence intervals were calculated using the Agresti–Coull method [27].

To further assess the indirect effect of community-level vaccination in the post-vaccination era, we estimated within-arm seroprevalence ratios (PRs) comparing the post- to pre-vaccination HPV-type-specific seroprevalence (for HPV types 16, 18, 31, 33, 35, 45, and 16/18 combined) using a log binomial generalized estimating equation (GEE) model to take account of within-arm clustering. HPV-type-specific seroprevalence was not directly compared between the arms as stated in the pre-analysis plan (S2 Text), as statistically significant differences were found between the arms at baseline prior to any HPV vaccination. To take account of possible

confounding, all estimates were adjusted for community-level self-reported maternal smoking, as a surrogate of community-level risk-taking behaviors. To investigate the effect of core group membership (a possible effect modifier) on the indirect effect, the secondary outcome of the study, we stratified the estimates by HSV-2 seropositivity.

To estimate the overall herd effect (the indirect effect) of gender-neutral and girls-only HPV vaccination compared to the counterfactual scenario, we further calculated the between-arm ratio of PRs, comparing the within-arm PR (adjusted for community-level maternal smoking) of Arm A or B (the intervention arms) to the PR of the control Arm C. The accompanying 95% confidence intervals were calculated according to the methodology of Altman and Bland [28].

Systematic outcome misclassification of the serological assay was quantified and corrected for assuming non-differential bias of the within- and between-arm estimates via probabilistic bias analysis [29]. Previously obtained estimates of test sensitivity and specificity were used at the outset (S1 Table) [30], assuming a constant probability distribution. If these prior estimates proved incompatible with the observed data, then a uniform probability density ranging from 0 to 1 was specified, to obtain all plausible values of the sensitivity or specificity compatible with the observed data. The resultant range of plausible values for the given HPV-type-specific sensitivity or specificity was then assumed, with a uniform probability density ranging from the given minimum to maximum value. The results from this sensitivity analysis were then used to quantify misclassification in the primary analysis.

All statistical analyses were conducted using the R statistical software package (version 3.6.0.).

## Results

### Baseline characteristics of the study population

In total, all 8,022 HPV-unvaccinated pregnant females under the age of 23 years who were resident in one of the 33 trial communities and had been invited to donate a blood sample to the FMC between the years 2005 and 2016 were identified. An additional 3,498 females were initially found to be ineligible due to being HPV vaccinated. In total, 4,007 participants were from the era preceding completion of vaccination (2005–2010), and 4,015 were from the post-vaccination era (2011–2016). Participants were excluded from the statistical analyses owing to HPV vaccination ($N = 49$) or being aged >22 years at sample donation ($N = 436$). In total, 7,531 women were included: 1,322, 1,289, and 1,304 from the pre-vaccination-era Arm A, B, and C communities, respectively, and 1,247, 1,158, and 1,211 from the same post-vaccination-era communities (Fig 2). The intracluster correlation coefficient was consistently low, at 0.007 for HPV16/18 seropositivity and 0.005 for HPV16/18/31/33/35/45 seropositivity (S2 Table).

The participants' age distributions in the pre-vaccination and post-vaccination eras were comparable, with the majority being 18 to 22 years old (S3 Table). The HSV-2 seroprevalence was materially similar between the arms, but somewhat higher in the pre-vaccination era as compared to the post-vaccination era (17.8% and 15.0%, respectively) (Fig 3). Community-level self-reported smoking was consistently higher in the control Arm C communities than in the gender-neutral vaccination Arm A and girls-only vaccination Arm B communities (S3 Table). The community-specific vaccination coverage among the eligible female birth cohorts for this study was negligible in the pre-vaccination era, from 2005 until 2010, and increased in the post-vaccination era in the intervention arm communities (from 5.6% to 52.5% in Arm A, and from 6.3% to 46.7% in Arm B) (Fig 4).

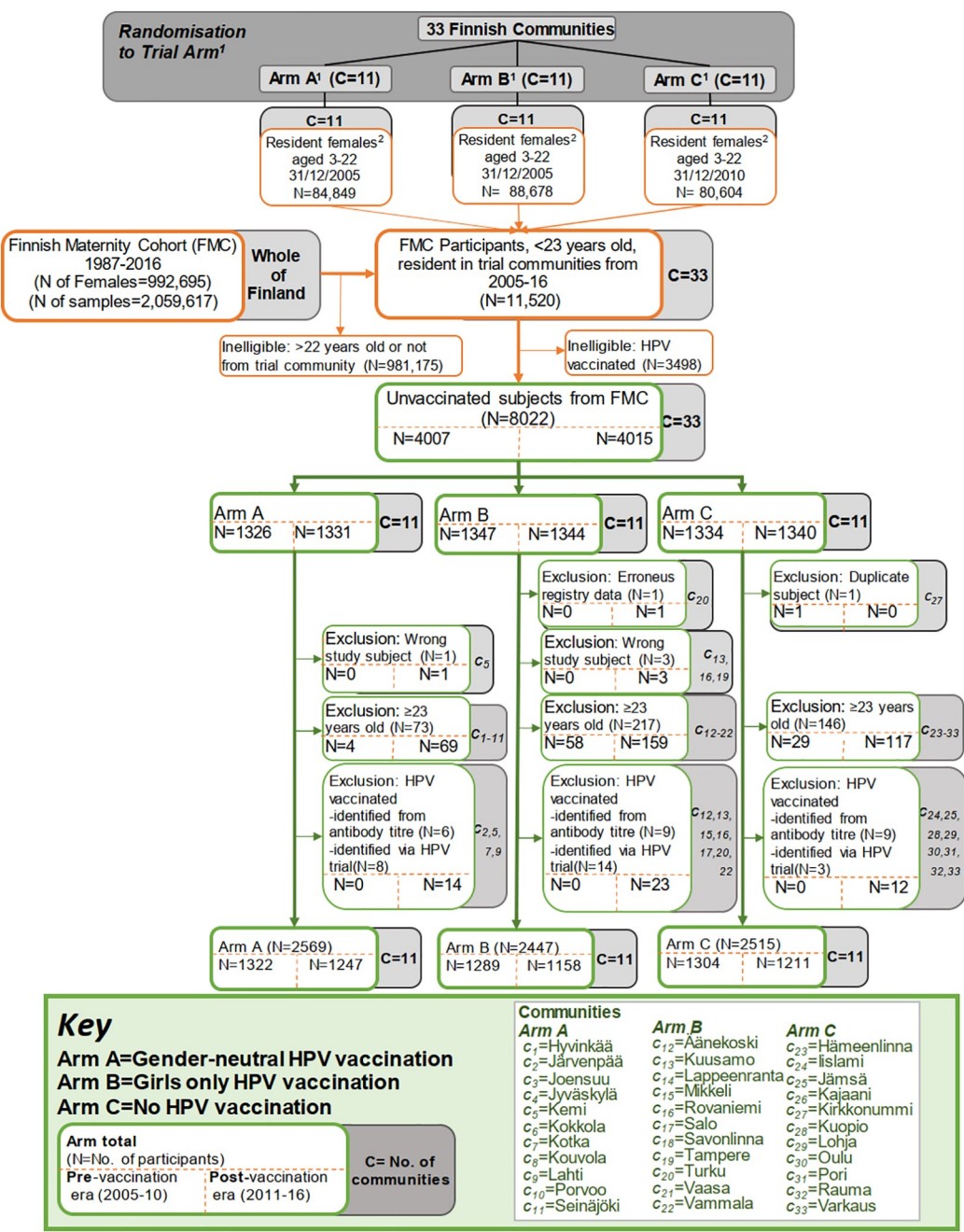

**Fig 2. Flow chart of the cross-sectional cohort study nested in the Finnish community randomized human papillomavirus (HPV) vaccination trial with stepwise subsequent exclusions.** [1]The arms are the trial arms from the cluster (community) randomized trial of HPV vaccination strategy, conducted in 2007–2010.[2]Includes all females aged 3–22 years who were resident in the communities specified as of the 31 December 2005 (data extracted from Statistics Finland).

## HPV seroprevalence by vaccination era

In the pre-vaccination era, HPV16/18 seroprevalence was high: 29.7%, 29.6%, and 26.8%, respectively, in the Arm A, B, and C communities. In the post-vaccination era, HPV16/18 seroprevalence was somewhat decreased (23.6%) in the gender-neutral vaccination Arm A

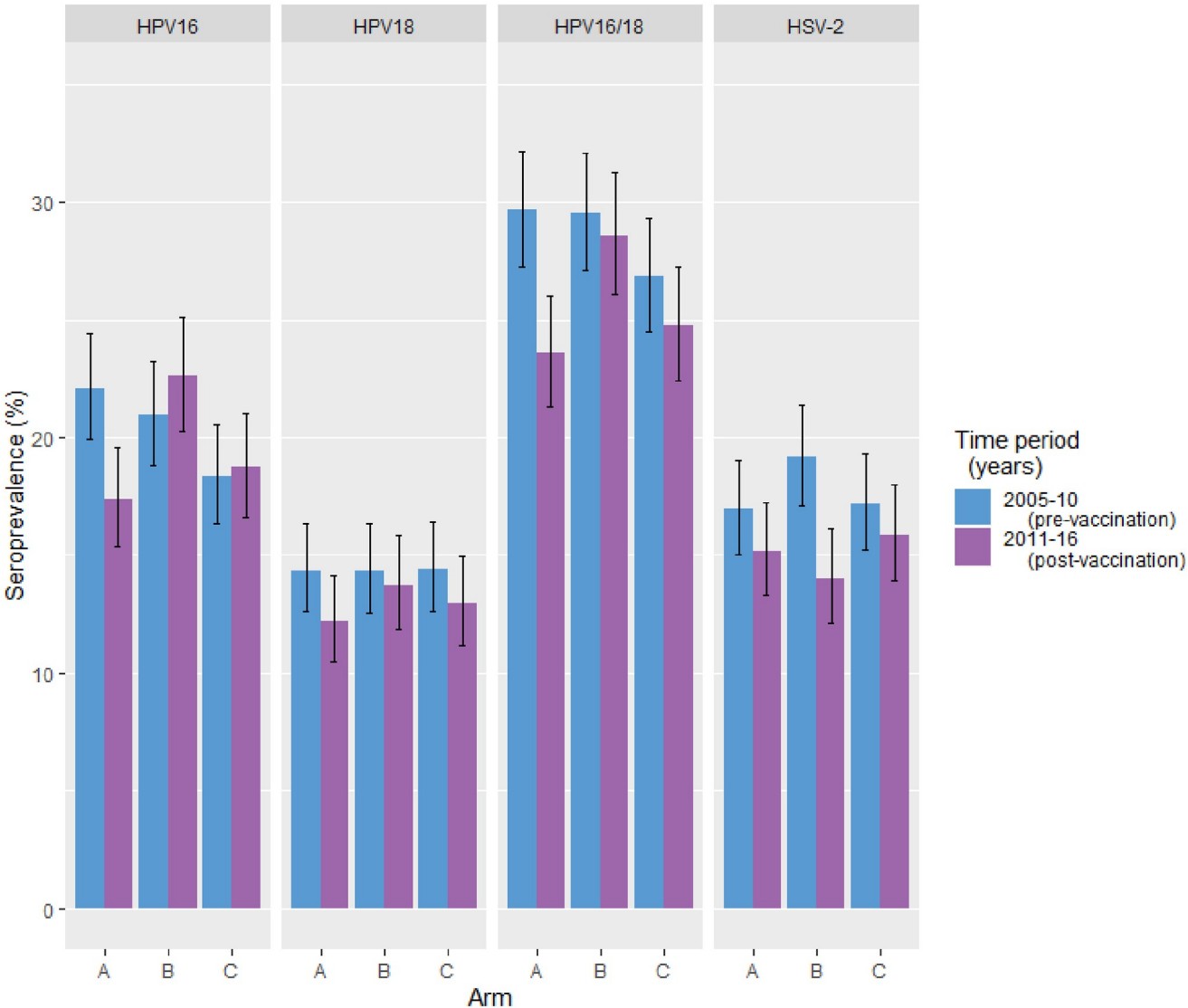

**Fig 3. Type-specific human papillomavirus (HPV) and herpes simplex virus type 2 (HSV-2) seroprevalence (%) among unvaccinated females under the age of 23 years by intervention strategy: Gender-neutral vaccination (Arm A), girls-only vaccination (Arm B), and control vaccination (Arm C).** Type-specific seroprevalence is stratified by time period of sample donation (pre-vaccination era, 2005–2010; post-vaccination era, 2011–2016).

communities (Fig 3; S4 Table). Notably, the HPV16 seroprevalence was decreased in the Arm A communities in the post-vaccination era compared to the pre-vaccination era (17.4% versus 22.1%). No decrease in HPV16 seroprevalence was noted in the girls-only vaccination Arm B or control C communities (Fig 3; S4 Table).

## Within-arm post- versus pre-vaccination-era HPV PRs

The within-arm HPV16/18 PR comparing the post- to the pre-vaccination era was notably decreased in Arm A. The HPV16/18 estimate was significantly somewhat decreased in the gender-neutral vaccination Arm A ($PR_{16/18}$ = 0.80, 95% CI 0.74–0.87), whereas in the girls-only vaccination Arm B and control Arm C, no significant reductions were noted ($PR_{16/18}$ = 0.98,

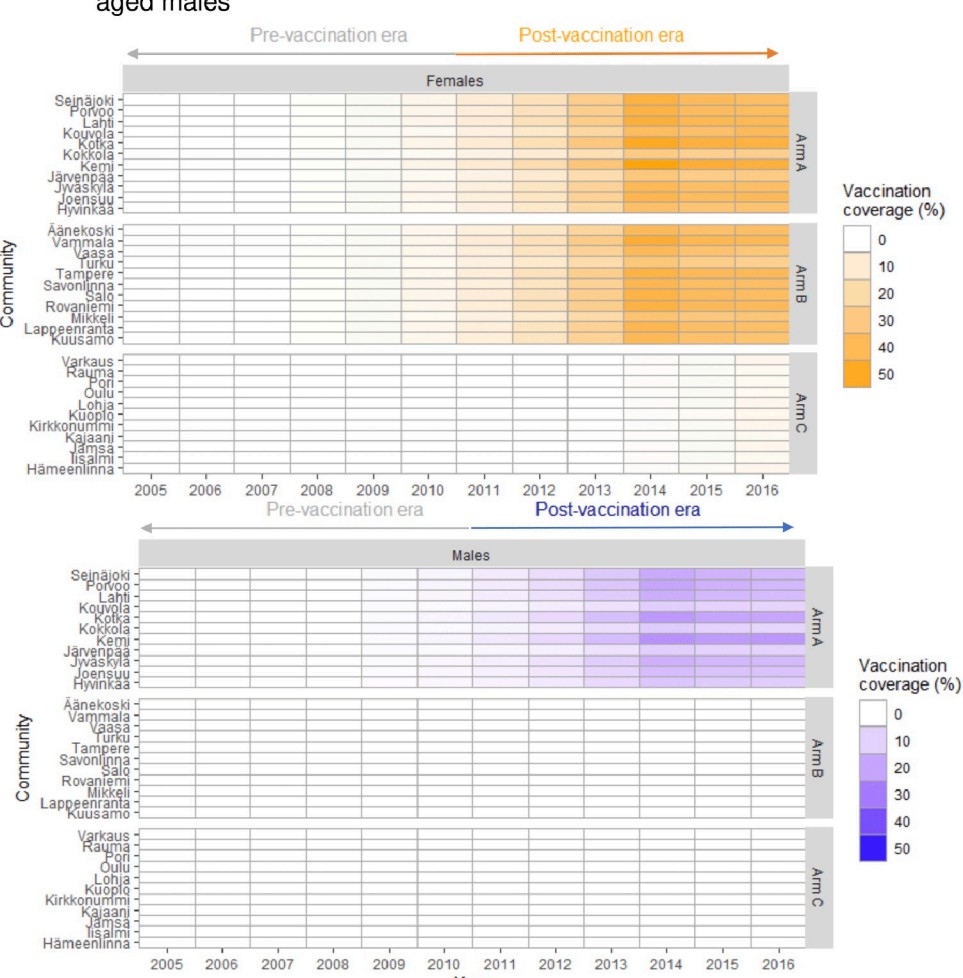

**Fig 4. Evaluation of human papillomavirus (HPV) vaccination coverage in the study population: Community-specific birth-cohort-weighted vaccination coverage of the consecutive community-randomized trial and national girls-only vaccination program.** Exposure to the indirect effects of HPV16/18 vaccination is defined as residing at the time of sample donation in one of the community-randomized HPV vaccination trial communities. Each row represents a trial community, and each column a year of the follow-up period. The community-specific vaccination coverage is calculated for pregnant females under the age of 23 years and includes vaccination of 12- to 15-year-old males and females in 2007–2010, and the national girls-only vaccination program launched in late 2013.

95% CI 0.85–1.12, in Arm B; $PR_{16/18}$ = 0.91, 95% CI 0.81–1.03, in Arm C) (Table 1). The HPV16 PR specifically was decreased in Arm A ($PR_{16}$ = 0.79, 95% CI 0.72–0.87). No corresponding decrease was observed in Arms B or C ($PR_{16}$ = 1.09, 95% CI 0.91–1.32, in Arm B; $PR_{16}$ = 1.01, 95% CI 0.86–1.20, in Arm C) (Table 1). After applying probabilistic bias analysis to correct for outcome misclassification, the within-arm HPV16/18 and HPV16 PR estimates in Arm A were found to be further decreased ($PR_{16/18}$ = 0.66, 95% CI 0.10–0.85, and $PR_{16}$ = 0.64, 95% CI 0.09–0.86, respectively). Also, the within-arm PR estimate for HPV18 was significantly decreased in the gender-neutral vaccination Arm A after accounting for the error due to outcome misclassification ($PR_{18}$ = 0.72, 95% CI 0.21–0.96) (Table 1).

**Table 1. Post- versus pre-vaccination HPV-type-specific adjusted seroprevalence ratio (PR) among unvaccinated Finnish females aged under 23 years.**

| HPV type | Post- versus pre-vaccination-era PR (95% CI) | | |
|---|---|---|---|
| | Arm A ($N = 1,247$ versus 1,322) | Arm B ($N = 1,158$ versus 1,289) | Arm C ($N = 1,211$ versus 1,304) |
| **Accounting for random error** | | | |
| 16 | 0.79 (0.72–0.87) | 1.09 (0.91–1.32) | 1.01 (0.86–1.20) |
| 18 | 0.86 (0.70–1.06) | 0.96 (0.74–1.24) | 0.89 (0.70–1.13) |
| 16/18 | 0.80 (0.74–0.87) | 0.98 (0.85–1.12) | 0.91 (0.81–1.03) |
| 31 | 0.90 (0.79–1.01) | 0.86 (0.73–1.02) | 0.72 (0.57–0.91) |
| 33 | 1.05 (0.88–1.26) | 0.94 (0.77–1.14) | 0.81 (0.63–1.03) |
| 35 | 0.70 (0.52–0.94) | 0.98 (0.64–1.51) | 0.77 (0.58–1.01) |
| 45 | 0.89 (0.69–1.14) | 1.01 (0.76–1.36) | 0.90 (0.64–1.26) |
| **Accounting for random error and systematic error** | | | |
| 16 | 0.64 (0.09–0.86) | 1.19 (0.98–3.70) | 1.07 (0.89–1.85) |
| 18 | 0.72 (0.21–0.96) | 0.89 (0.39–1.12) | 0.79 (0.21–1.03) |
| 16/18 | 0.66 (0.10–0.85) | 0.92 (0.44–1.07) | 0.84 (0.24–1.01) |
| 31 | 0.79 (0.20–1.00) | 0.75 (0.15–0.97) | 0.55 (0.07–0.78) |
| 33 | 1.14 (0.86–2.73) | 0.85 (0.31–1.13) | 0.66 (0.15–0.94) |
| 35 | 0.52 (0.07–0.83) | 0.90 (0.44–1.25) | 0.59 (0.10–0.91) |
| 45 | 0.73 (0.19–1.06) | 1.01 (0.78–1.33) | 0.79 (0.25–1.13) |

Comparisons are between 2 time periods of sample donation (2011–2016, post-vaccination era, versus 2005–2010, pre-vaccination era), stratified by intervention Arm A (gender-neutral HPV vaccination), Arm B (girls-only HPV vaccination), and Arm C (control vaccination), accounting for random error and accounting for random error and systematic error due to outcome misclassification. The estimates corrected for random error only are adjusted for community-level smoking. Corresponding unadjusted estimates are displayed in S5 Table.

The HPV35 PR estimate was significantly decreased in the gender-neutral vaccination Arm A ($PR_{35} = 0.70$, 95% CI 0.52–0.94; Table 1). However, this finding appeared to be essentially replicated in the control Arm C ($PR_{35} = 0.77$, 95% CI 0.58–1.01; Table 1). No decrease in HPV35 seroprevalence was observed in the girls-only vaccination Arm B ($PR_{35} = 0.98$, 95% CI 0.64–1.51). HPV31 was non-significantly slightly decreased in both the gender-neutral vaccination Arm A and the girls-only vaccination Arm B ($PR_{31} = 0.90$, 95% CI 0.79–1.01, in Arm A; $PR_{31} = 0.86$, 95% CI 0.73–1.02, in Arm B), but was further decreased in the control Arm C with no HPV vaccination ($PR_{31} = 0.72$, 95% CI 0.57–0.91). HPV33 was not decreased in Arm A ($PR_{33} = 1.05$, 95% CI 0.88–1.26), while it approximately stayed the same in Arm B ($PR_{33} = 0.94$, 95% CI 0.77–1.14) and non-significantly decreased slightly in Arm C ($PR_{33} = 0.81$, 95% CI 0.63–1.03). On the other hand, HPV45 was non-significantly marginally decreased in Arm A ($PR_{45} = 0.89$, 95% CI 0.69–1.14), approximated the null in Arm B ($PR_{45} = 1.01$, 95% CI 0.76–1.36), and in Arm C was decreased in a similar manner as in Arm A ($PR_{45} = 0.90$, 95% CI 0.64–1.26) (Table 1).

The HPV16 PR estimate was also noticeably decreased among the HSV-2 seropositive individuals in the gender-neutral vaccination Arm A ($PR_{16} = 0.64$, 95% CI 0.50–0.81). Most estimates for vaccine-protected HPV types were also decreased among the HSV-2 seropositive individuals, especially in Arm A ($PR_{31} = 0.74$, 95% CI 0.53–1.02; $PR_{35} = 0.57$, 95% CI 0.37–0.88; $PR_{45} = 0.64$, 95% CI 0.37–1.08), albeit sometimes with borderline statistical significance. The findings for HPV31 and HPV45 were, however, replicated in the control Arm C ($PR_{31} = 0.64$, 95% CI 0.42–0.98; $PR_{45} = 0.69$, 95% CI 0.37–1.30) (S6 Table).

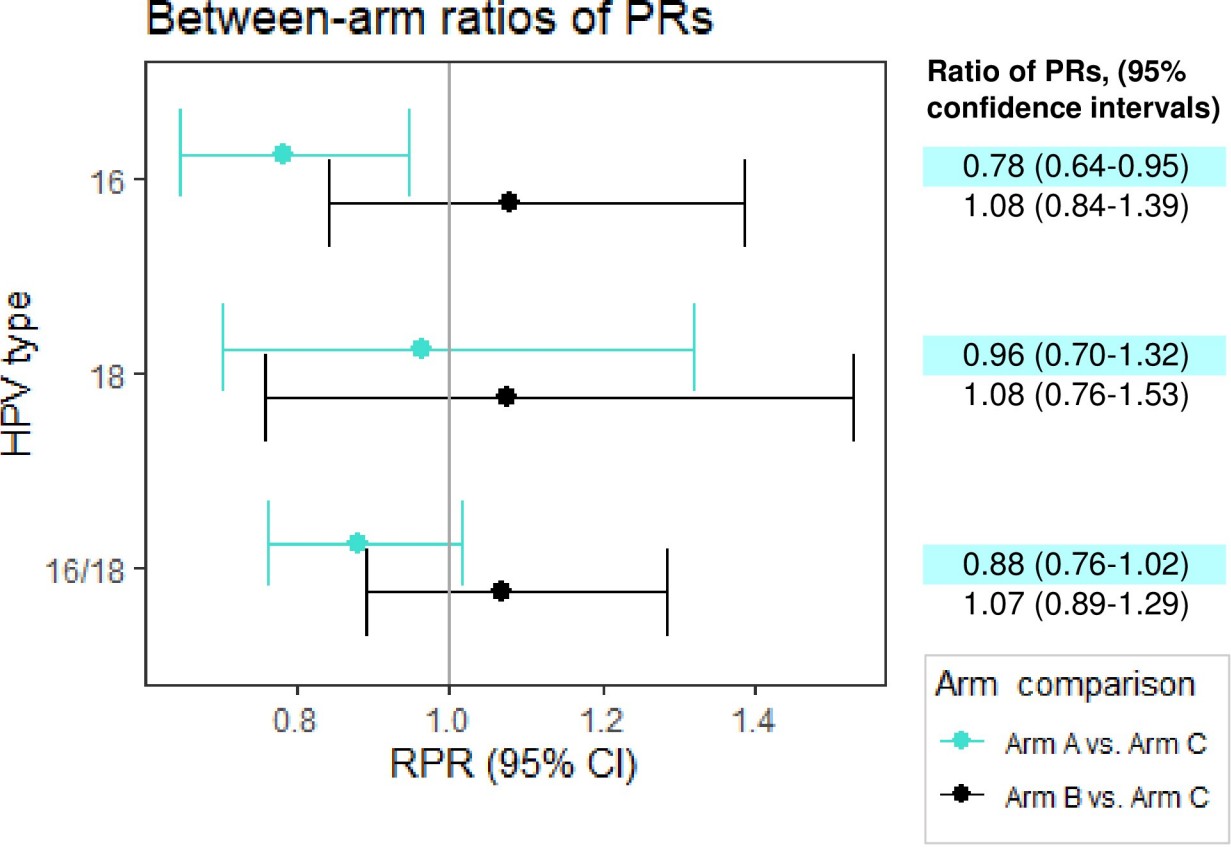

**Fig 5. Ratio of human papillomavirus (HPV) seroprevalence ratios (PRs) comparing Arm A/B to Arm C.** Arm-specific PRs comprise post-vaccination to pre-vaccination-era HPV PRs among pregnant unvaccinated Finnish females, aged under 23 years, and adjusted for community-level maternal smoking. RPR, ratio of seroprevalence ratios.

### Between-arm comparison of the post- versus pre-vaccination-era PRs

To account for possible secular trends, between-arm comparisons of the within-arm post- versus pre-vaccination-era PRs were made, comparing the ratios from HPV vaccination arms to the ratios from the control Arm C. The HPV16 ratio of PRs (RPR), remained decreased when comparing the gender-neutral vaccination Arm A to the control Arm C (RPR$_{16}$ = 0.78, 95% CI 0.64–0.95) (Fig 5).

### Discussion

We nested a cross-sectional cohort within a population-based, community-randomized HPV16/18 vaccination trial to estimate changes over time in HPV16/18 seroprevalence created by gender-neutral or girls-only vaccination strategies, using pre- and post-vaccination-era sera from unvaccinated women resident in the trial communities. The HPV16 and HPV18 sero-prevalence was somewhat decreased in young unvaccinated women after gender-neutral vacci-nation. This was observed although the vaccination coverage was only moderate to low. Most importantly, a degree of partial herd effect against HPV16 was observed over time within the gender-neutral vaccination arm, when compared to the counterfactual control arm, and within the HSV-2 seropositive core group, representing those with high contact rate. Girls-only HPV

vaccination with moderate vaccination coverage did not result in any notable HPV16 herd effect.

The level of vaccination coverage required for herd effect is a function of a given HPV type's basic reproduction number. This in turn is a function of the effective transmission rate and the mean duration of infection, which for HPV16 is especially long. Thus, the vaccination coverage required to achieve herd effect against HPV16 is expected to be high, higher than for other HPV types [14], and for girls-only vaccination this indeed seems to be the case [16]. Furthermore, the predicted herd effect is 25% to 50% greater for a gender-neutral than for a girls-only vaccination scenario [23]. Our study provides empirical evidence that when vaccination coverage is suboptimal, a gender-neutral vaccination strategy optimizes HPV16 herd effect and thus effectiveness of vaccination.

This observation is of major importance, if the call for action to eliminate cervical cancer is realistically going to be achieved. The gender-neutral vaccination strategy, with its sturdier impact on both HPV16 and HPV18, may assist in overcoming the obstacle of suboptimal girls-only vaccination coverage.

Apart from HPV16 being the most oncogenic HPV type, both dynamic transmission models and randomized trials have suggested that HPV16 is the most difficult to achieve herd protection against [14,17]. The vaccination coverage required to achieve herd protection against a given vaccine-protected type in addition to being strategy-dependent is also population-dependent [17]. The observation of a degree of partial HPV16 herd effect also in the core group following gender-neutral vaccination is reassuring, since modeling studies have suggested that the existence of the core group defies creation of herd effect [17]. When using PCR for the determination of current HPV infections, and concomitant *Chlamydia trachomatis* infection as a proxy of sexual risk-taking behavior, only HPV18 herd effect has been observed among the core group [31]. Whereas the seroprevalence comparisons documented a HPV16 herd effect in those with 'ever' core group membership, with HSV-2 seropositivity as the proxy. Thus, implementing a gender-neutral vaccination strategy is likely to deliver also on targets of equity in eliminating cervical cancer.

Previously, when following up the 1992–1995 birth cohorts of our community-randomized trial, no evidence of HPV16 herd effect among unvaccinated females was observed with PCR-defined endpoints [13]. DNA positivity as identified by one-time PCR-positivity is not a measure of cumulative infection and is also subject to outcome misclassification owing to its inability to distinguish persistent infections from transient depositions [32]. The resulting suboptimal specificity probably biased the previous estimates of HPV16/18 herd effect [13], and likely resulted in an underestimation of the true effects.

Sensitivity analyses assuming that serology has imperfect sensitivity to identify cumulative HPV exposure found that the degree of misclassification was HPV type specific. Furthermore, the previous validation methods [30] may have underestimated the true specificity. The remaining misclassification biased the estimates towards the null point. The previously reported sensitivity estimates were particularly low for HPV18. After quantifying and correcting for this bias, the HPV18 PR estimates were in line with earlier PCR-based observations of HPV18 herd effect after gender-neutral vaccination [13].

This study is limited by the imperfect ability of HPV serology to identify all cumulative HPV infection. However, HPV antibodies are a measure of persistent infection, thus by identifying cumulative infection by seroconversion, we identify women with true persistent HPV infection and exclude the apparent issue of the absence of HPV seroconversion in a proportion of women who have had only an HPV deposition.

Earlier studies were restricted to the birth cohorts that participated in the community-randomized trial [13]. We now also included females from the unvaccinated birth cohorts, 1996–

1997, subsequent to the trial cohorts. Because HPV is a sexually transmitted virus, HPV transmission moves in the direction of the older to younger birth cohorts. Thus, the herd effect was expected to be stronger in the younger vaccinated cohorts [14]. Probably the older vaccinated male and female 1992–1995 birth cohorts have conferred indirect protection to the subsequent birth cohorts in the gender-neutral vaccination arm, due to the disruption in HPV transmission [33].

This is to our knowledge the only serosurvey among unvaccinated females following a community-randomized trial of different vaccination strategies. A previous serosurvey conducted in Australia among unvaccinated males to evaluate the first-order herd effect of girls-only vaccination found somewhat decreased HPV16/18 seropositivity in its post-vaccination era [34]. However, males seroconvert following HPV infections at lower rates than females, thus resulting in a possible underestimation of the true reduction in infection prevalence [35].

This study is strengthened by its utilization of the Finnish infrastructure of population-based intervention cohorts, biobanks, and registries, linkable via unique personal identification numbers. Our extraction of serum samples from all eligible participants in the population-based FMC [21] provided a sufficient population-representative sample size to evaluate vaccination-strategy-specific differences in HPV seroprevalence and account for random error. Further to this, our sampling of only pregnant females under 23 years at the time of sample donation captures the age distribution at which the HPV incidence curve peaks and the demographic most at risk.

This study may be limited in its generalizability, as the average age of mothers at first live birth in Finland is approximately 29 years [36]. It is possible that our study population of pregnant females under 23 years have above-average sexual risk-taking behaviors and lifetime risk of acquiring HPV. Our results may have incomplete transportability to populations that have differing baseline risk and sexual network structures This study is also limited to the setting of moderate vaccination coverage; therefore, the findings are not generalizable to scenarios with greater vaccination coverage. Furthermore, the findings may also have incomplete generalizability to a scenario with more consecutively vaccinated birth cohorts and a longer period of time between vaccination initiation and follow-up. Given our inclusion criteria in this study, it is also possible that participants may have moved between communities between the commencement of sexual activity and sample donation, which may introduce some bias in our estimates. However, given that the participants were all pregnant females under the age of 23 years, it is possible that they are the portion of source population that are least likely to move to a new community.

It may also be possible that changes over time in the sexual network or risk-taking behavior have altered HPV seroprevalence even in the absence of vaccination. The observed changes in HSV-2 seroprevalence could conceivably be interpreted as evidence of this. However, HSV-2 epidemiology has globally been undergoing complex changes in recent decades, with HSV-1 increasing and HSV-2 decreasing as the main cause of genital herpes infections [37]. Therefore, the observed decrease in HSV-2 seropositivity over time is not entirely unexpected and may be independent of any changes in HPV incidence over the time frame. However, with respect to this, our study is strengthened by its design, as the within-arm seroprevalence comparisons in Arm C, where no HPV vaccination was applied, provided us with a counterfactual estimate to tackle such possible secular trends.

Our results suggest that when HPV vaccination coverage is moderate, only gender-neutral vaccination establishes herd effect against HPV16 and HPV18 among unvaccinated females. This finding supports the implementation of a gender-neutral HPV vaccination policy to achieve optimal vaccine effectiveness when obtaining a girls-only vaccination coverage of 90% is impossible.

## Supporting information

**S1 Checklist. STROBE checklist.**
(DOCX)

**S1 Fig. Lexis diagrams depicting the vaccinated cohorts and vaccination coverage among the eligible birth cohorts of the study population, by arm and gender.** (a) Among females; (b) among males.
(DOCX)

**S2 Fig. Type-specific human papillomavirus (HPV) seroprevalence (%) among unvaccinated females under the age of 23 years by vaccination strategy—gender-neutral vaccination (Arm A), girls-only vaccination (Arm B), and control vaccination (Arm C)—and time period of sample donation (pre-vaccination era, 2005–2010, and post-vaccination era, 2011–2016).**
(DOCX)

**S3 Fig. Ratio of HPV seroprevalence ratios (RPR) comparing Arm A/B to Arm C.** Arm-specific PRs comprise post-vaccination to pre-vaccination-era HPV seroprevalence ratios among pregnant unvaccinated Finnish females aged under 23 years old, and are adjusted for community-level maternal smoking.
(DOCX)

**S1 Table. Sensitivity and specificity parameters of pseudovirion-based serology in measuring cumulative HPV exposure used in the probabilistic bias analysis.**
(DOCX)

**S2 Table. Intracluster correlation coefficient (ICC) of any human papillomavirus (HPV) type seropositivity among pregnant females donating sera during the pre-vaccination era, 2005–2010.** HSV-2, herpes simplex virus type 2.
(DOCX)

**S3 Table. Characteristics of the study population after exclusions owing to ineligibility.**
(DOCX)

**S4 Table. Absolute HPV-type-specific seroprevalence among unvaccinated pregnant Finnish women stratified by trial arm and vaccination era (2005–2010 is defined as the "pre-vaccination era" and 2011–2016 as the "post-vaccination era"), and additionally by HSV-2 seropositivity.**
(DOCX)

**S5 Table. Unadjusted HPV-type-specific seroprevalence ratio (PR) among unvaccinated Finnish females comparing the post-vaccination era to the pre-vaccination era.** Comparisons are between 2 time periods of sample donation (2011–2016, post-vaccination era, versus 2005–2010, pre-vaccination era), stratified by intervention Arm A (gender-neutral HPV vaccination), Arm B (girls-only HPV vaccination), and Arm C (control vaccination).
(DOCX)

**S6 Table. Adjusted seroprevalence ratio (PR) of HPV seropositivity by HPV type among pregnant, unvaccinated Finnish females under the age of 23 years by study arm (gender-neutral vaccination Arm A, girls-only vaccination Arm B, or control Arm C), comparing time period of sample donation (post-vaccination era, 2011–2016, compared to the pre-vaccination era, 2005–2010), and stratified by herpes simplex virus type 2 serostatus.** All

estimates are adjusted for smoking. na, not available.
(DOCX)

**S1 Text. Supplementary methods (laboratory analysis and statistical analysis).**
(DOCX)

**S2 Text. Prospective pre-analysis plan.**
(PDF)

**S3 Text. Trial protocol and report analysis plan (HPV-040 trial).**
(PDF)

## Acknowledgments

The authors wish to thank the steering committee of the HPV-040 trial—Allan Donner, Eduardo Franco, Pauli Leinikki, Achim Schneider, and Margaret Stanley—for all their scientific advice and support throughout the study. In addition, they would like to thank Kat French for generating the random allocation sequence of the original community-randomized trial. The authors also wish to thank Sara Kuusiniemi and Indira Adhikari for their part in the collection and handling of the serum samples used in this study from the larger FMC biobank, and Mika Gissler for providing data on self-reported maternal smoking, collected as part of the Finnish Medical Birth Register.

## Author Contributions

**Conceptualization:** Penelope Gray, Tapio Luostarinen, Simopekka Vänskä, Heljä-Marja Surcel, Helena Faust, Joakim Dillner, Matti Lehtinen.

**Data curation:** Penelope Gray, Hanna Kann, Tiina Eriksson, Tapio Luostarinen, Heljä-Marja Surcel.

**Formal analysis:** Penelope Gray, Hanna Kann.

**Funding acquisition:** Penelope Gray, Joakim Dillner, Matti Lehtinen.

**Investigation:** Penelope Gray, Helena Faust, Joakim Dillner, Matti Lehtinen.

**Methodology:** Penelope Gray, Hanna Kann, Ville N. Pimenoff, Tapio Luostarinen, Simopekka Vänskä, Heljä-Marja Surcel, Helena Faust, Joakim Dillner, Matti Lehtinen.

**Project administration:** Penelope Gray, Tiina Eriksson, Tapio Luostarinen, Heljä-Marja Surcel, Helena Faust, Matti Lehtinen.

**Resources:** Penelope Gray, Tiina Eriksson, Tapio Luostarinen, Heljä-Marja Surcel, Helena Faust, Joakim Dillner, Matti Lehtinen.

**Software:** Penelope Gray, Matti Lehtinen.

**Supervision:** Ville N. Pimenoff, Simopekka Vänskä, Heljä-Marja Surcel, Helena Faust, Joakim Dillner, Matti Lehtinen.

**Validation:** Penelope Gray, Hanna Kann, Helena Faust, Matti Lehtinen.

**Visualization:** Penelope Gray.

**Writing – original draft:** Penelope Gray, Ville N. Pimenoff, Matti Lehtinen.

**Writing – review & editing:** Penelope Gray, Hanna Kann, Ville N. Pimenoff, Tiina Eriksson, Tapio Luostarinen, Simopekka Vänskä, Heljä-Marja Surcel, Helena Faust, Joakim Dillner, Matti Lehtinen.

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
