## [Editor Report · Decision Letter 0]

15 Jan 2020

Dear Dr Gray, 

Thank you for submitting your manuscript entitled "Gender-neutral vaccination is the key to HPV16 herd effect when vaccination coverage is limited: a community-randomised trial of vaccination strategy" for consideration by PLOS Medicine.

Your manuscript has now been evaluated by the PLOS Medicine editorial staff [as well as by an academic editor with relevant expertise] and I am writing to let you know that we would like to send your submission out for external peer review.

**Please be aware that, due to the voluntary nature of our reviewers and academic editors, manuscript assessment may be subject to delays during the holiday season. Thank you for your patience.**

Kind regards,

Adya Misra, PhD,

Senior Editor

PLOS Medicine

---

## [Decision Letter · Decision Letter 1]

13 May 2020

Dear Dr. Gray,

Thank you very much for submitting your manuscript "Gender-neutral vaccination is the key to HPV16 herd effect when vaccination coverage is limited: a community-randomised trial of vaccination strategy" (PMEDICINE-D-20-00112R1) for consideration at PLOS Medicine. 

[LINK]

In light of these reviews, I am afraid that we will not be able to accept the manuscript for publication in the journal in its current form, but we would like to consider a revised version that addresses the reviewers' and editors' comments. Obviously we cannot make any decision about publication until we have seen the revised manuscript and your response, and we plan to seek re-review by one or more of the reviewers. 

We expect to receive your revised manuscript by Jun 03 2020 11:59PM. Please email us (plosmedicine@plos.org) if you have any questions or concerns.

We look forward to receiving your revised manuscript. 

Sincerely,

Emma Veitch, PhD

PLOS Medicine

On behalf of Clare Stone, PhD, Acting Chief Editor,

PLOS Medicine

plosmedicine.org

*In the title and throughout the paper, the term "community RCT" is used, do the authors mean "cluster RCT" which might be the more commonly recognised term? (in that randomization is of groups of individuals, not individuals themselves). The authors could consider using cluster RCT instead, this might be more generally understood and recognised.

*Although the authors use CONSORT to aid reporting of the trial, it would be good to explicitly mention this in the methods (and call out there the supporting file which basically constitutes the CONSORT checklist), and also to consider using the specialised CONSORT extension for cluster RCTs (http://www.consort-statement.org/extensions?ContentWidgetId=554) as this contains specific advice for particular elements of design and analysis that should be elaborated in more detail for a cluster RCT.

*In the abstract, it would be good to state how many clusters were in each randomized arm in the trial. 

*In the last sentence of the Abstract Methods and Findings section, it would be good to include a brief description of any key limitation(s) of the study's design/methods.

*At this stage, we ask that you include a short, non-technical Author Summary of your research to make findings accessible to a wide audience that includes both scientists and non-scientists. The Author Summary should immediately follow the Abstract in your revised manuscript. This text is subject to editorial change and should be distinct from the scientific abstract. Please see our author guidelines for more information: https://journals.plos.org/plosmedicine/s/revising-your-manuscript#loc-author-summary

*If possible, please tweak the in-text referencing style to match the PLOS Medicine format (numerals in square brackets, ie [1, 2]) - if using referencing software this should be fairly straight forward.

*Per the reviewers' comments, it would be good to explain in more detail in the manuscript and acknowledge the degree of novelty over and above prior studies also examining this question. 

*In the Methods, it would be good to spell out the randomization procedures more clearly, per the CONSORT statement (either standard or cluster CONSORT should explain what needs to be detailed), but specifically an explanation of how allocation to arms was concealed and so on. 

Comments from the reviewers:

Reviewer #1: "Gender-neutral vaccination is the key to HPV16 herd effect when vaccination coverage is limited: a community-randomised trial of vaccination strategy" investigates the effects of moderate gender-neutral vs. girls-only vs. control (none) human papillomavirus (HPV), through a community-randomized trial in Finland, on herd immunity. The main analysis involving post- vs. pre-vaccination HPV prevalence/correlation was performed on 8,022 qualifying unvaccinated pregnent women whom were 23 years old or under and resident in the 33 involved communities, between 2005 and 2016. It was concluded that only gender-neutral vaccination results in herd protection against HPV16/18 (and not girls-only or no vaccination), when vaccination coverage is moderate.

The major strengths of this study lie in the original large-scale community-randomized design, and the statistical analysis attempting to take linked attributes (e.g. maternal smoking, HSV-2 seropositivity) into account. The general findings are also in accordance with expectations and modelling attempts (e.g. [12]), given the primary spread of HPV through intercourse. However, there are a number of key concerns:

1. Firstly, relatively similar analyses (comparing gender-neutral versus girls-only impact on HPV prevalence) on the same data have been published by some of the same authors (i.e. citations [16],[8]). In fact, some of the details available in these prior work (e.g. on the nature of the 33 communities, with each community having at least 35,000 inhabitants, and being at least 35km from each other) would probably be appropriately included as supplementary material here. In any case, the same broad conclusions had been reached in [16]: "In conclusion, while gender-neutral strategy enhanced the effectiveness of HPV vaccination for cross-protected HPV types with low to moderate coverage, high coverage in males appears to be key to providing a substantial public health benefit also to unvaccinated females".

The immediate reaction would then be on the additional contributions of this manuscript, on top of and over these previous works ([16],[8]). From what could be understood, the main distinction was that cervicovaginal samples taken at 18.5 years from all (vaccinated/unvaccinated) female participants were used for PCR analysis in [16], as opposed to seroprevalance amongst unvaccinated pregnant females at or under 23 years only - but also from non-participating cohorts - for this study. In general, the authors might consider stressing the presence of their directly-relevant previous work, and describe more precisely the relative strengths/weaknesses of this paper in comparison to those previous works.

2. While the authors do mention that "one-time PCR positivity is not a measure of cumulative infection", the discussion then goes on to admit the imperfect sensitivity of serology (as a measure of persistent infection), and seemingly uses PCR-based observations as the comparison standard after correcting for serology bias. It seems clear that the sensitivity of HPV detection (from serology) can have an appreciable effect on the outcome of the analyses. As such, the authors might consider quantifying the degree of this imperfect sensitivity (e.g. against some [PCR?] gold standard?), and how exactly the serology analyses were "quantified and corrected" (perhaps as an addition to the supplementary Laboratory Analysis).

3. A major focus of the paper is on "herd protection", given its emphasis in the abstract's motivation and conclusions, in particular the statement that "...when vaccination coverage is moderate, only gender-neutral vaccination results in herd protection against both HPV types 16 and 18". The directly relevant results shown in Figure 4 do seem to suggest a statistically-significant reduction in seroprevalence of HPV (from ~30% to ~23%) for gender-neutral (Arm A) vaccination, compared to only a marginal reduction for girls-only (Arm B) vaccination. However, the question is whether such a reduction would qualify as "herd protection". The authors might consider discussing the criteria for herd protection to be validated (i.e. if the seroprevalence reduction were to say 27% instead of 23%, would there still be herd protection?)

Moreover, the results from Figure 4 appear to imply that the declaration in the Discussion section that "...the gender-neutral vaccination strategy with its superb impact on both HPV16 and 18 overcomes the obstacle of suboptimal girls-only vaccination coverage", might be slightly exaggerated.

4. Given the recommendation on aggregate population-level herd protection, it might be appropriate to have an idea of the movement dynamics of the study participants, if possible. In particular, did they tend to stay within their original communities (which determined the Arm they belonged to), or was movement between communities common? This is especially since it is stated that "...exposure to the indirect effects of HPV16/18 vaccination is defined as residing at the time of sample donation in one of the community-randomized HPV vaccination trial communities", which appears to implicitly allow for such movement. A brief discussion of how such movement (if prevalent) could inform the recommendations would be appreciated.

5. While the analyses were adjusted for maternal smoking, which is mentioned to be consistently higher in the control Arm C communities, this gives rise to the question of whether smoking is expected to have any direct effect on HPV/cervocal cancer prevalence. Moreover, how exactly were these statistical adjustments (including random/systematic error introduced in Table 1) implemented?

Also, since it is stated that "...all estimates were adjusted for community-wise self-reported smoking", the authors might comment whether omitting the adjustments produces similar conclusions in any case. This is especially since Figure 5 suggests that adjusting for smoking has caused Arm B (girls-only vaccination) to demonstrate higher HPV seroprevalence ratios compared to Arm C (the control)

6. The population numbers in Figure 2 may be a little off. In particular, pre-vaccination total N=4006 is stated at the beginning, but the sum of the three Arms in the next step, 1326+1347+1334 is 4007. Similarly, post-vaccination total N=4016, but the sum 1331+1344+1340 is 4015.

Additional minor comments follow:

7. It is stated in the Introduction that "Furthermore, in our community-randomized trial, gender-neutral HPV vaccination has provided significantly stronger herd effect against HPV18/31/33/35...". It might be clarified that this is from analysis performed in previously-published studies.

8. The HBV abbreviation seems to appear without an initial definition (hepatitis B); it might also be relevant to mention that the HBV vaccine shows no clinically-relevant interference in antibody response to HPV antigens, if true.

9. This sentence under the Statistical Analyses subsection might be reworded: "To evaluate the extent of exposure among the study population of pregnant females, where The exposure is defined as the indirect exposure to HPV16/18 vaccination due residing at the time of sample donation in one of the communities of the community-randomized trial".

10. For the sentence "Particularly the HPV16 seroprevalence was significantly decreased in the Arm A communities in the post-vaccination era compared to the pre-vaccination era (22.1% versus 17.4%)", the percentage figures might appear in the same order as the eras they reference (i.e. 17.4% versus 22.1%)

Reviewer #2: This is a carefully designed and appropriate study utilising HPV seroprevalence in young pregnant women ( plus some age matched males) from communities in Finland from pre and post HPV vaccination eras to deliver a randomised investigation to measure the potential impact of gender neutral vaccination to achieve better herd immunity The principle conclusion is that when HPV vaccination coverage is moderate, only gender-neutral vaccination establishes herd protection against HPV16 and HPV18, among unvaccinated females. This manuscript provides important support for the implementation of a gender-neutral HPV vaccination policy to achieve optimal vaccine effectiveness when obtaining a girls-only vaccination coverage of 90% is impossible. Previous studies have utilised HPV DNA PCR endpoints and have not seen such herd immunity impacts with lower coverage. This is discussed as likely due to this measure not reflecting cumulative infection and inability to distinguish persistent virus and transient infection.

Reviewer #3: In this study, Gray et al. use Finnish HPV serology data to examine whether gender-neutral HPV vaccination leads to more herd effects than girls-only vaccination in unvaccinated young women. The use of serology data presents a different and interesting way of assessing HPV exposure compared with most other herd effect studies which have used HPV DNA detection. Notable study strengths include the large sample size, high participant consent rate, the linkage of data with a large community-randomized HPV vaccination trial, and the inclusion of a sensitivity analysis to assess the likely impact of non-differential misclassification error. However, some notable limitations of the study include the low representativity of the sample (pregnant women <23) and limited ability to control for possible changes over time occurring in the background risk of HPV exposure (secular trends). While the topic is important and the study may offer some interesting insights, I have some major comments regarding the analysis and interpretations:

* The authors estimate that HPV vaccination coverage is approximately 47-53% in the eligible female birth cohorts in the post-vaccination era. However, very few women were excluded in the flowchart due to previous vaccination in the post-vaccination era (14/1331 in arm A, 23/1344 in arm B, and 12/1340 in arm C). This suggests that the women in the Finnish Maternity Cohort up to 2016 are a population of women where HPV vaccine coverage is at best 1-2% (likely they skew older and include women who may have been missed by the community trial). The authors should consider whether it is plausible that there would be herd effects when vaccination coverage appears to be so low in the study sample.

* The declines in HSV-2 seroprevalence over time suggest that there may be declines in HPV seroprevalence over time that may not be entirely attributable to vaccination programs, possibly due to differences in age distribution or in sexual behaviors. I would have liked to see this included somehow in the analyses, perhaps through the inclusion of a time trend in prevalence common to all HPV types. A time trend could have been included through the use of a difference in differences analysis, for example. Currently there is little mention of this in the discussion and of the possibility that observed changes could be due to other trends over time. 

* The study sample is meant to be restricted to unvaccinated women. We need more information on how the authors identified "HPV-vaccinated" women based on their antibody levels, this is not very clear currently.

Other comments:

* I personally find it problematic and stigmatizing to label HSV-2 seropositives as a "sexual risk-taking core group" given how common HSV-2 is in the general population. Please consider rephrasing this in the paper as women who are likely to have had higher exposure to STIs, which is less judgmental. 

* The authors claim there is little difference in age distribution pre- and post-vaccination. However, small differences in age could have a big impact on HPV seroprevalence given the large expected increase in HPV prevalence with age during the late teens. It would be helpful to give the mean age of women pre- and post-vaccination or include a table of the age distribution by era in the appendix. Is there any reason that age was not adjusted for in the log binomial model?

* The authors mention that pregnant women <23 are likely to be a more sexually active group, given that the average age at pregnancy is 29. The authors may want to mention that herd effects are expected to be lower and harder to detect in more sexually active groups, so their estimates are likely to underestimate herd effects in the general population.

* The serum samples were tested for more HPV types than were presented (39, 51, 52, 58, 59, 66, 68, 73). Why not present the results for these types, which are not expected to be affected by vaccine cross-protection? This would give further information as to whether there are time trends in HPV seroprevalence over time that are not attributable to vaccination.

Reviewer #4: This manuscript addresses the important topic of who to vaccinate against HPV. It is clearly written and makes an important contribution to our understanding of herd immunity. Herd protection was estimated via a repeated cross-sectional cohort study of all pregnant women under the age of 23 within thirty-three communities participating in a community-randomized HPV vaccination trial. Cumulative infection in unvaccinated patients was estimated using L1 VLP serology. Communities were then randomized to one of three trial Arms: Arm A, gender-neutral HPV vaccination; Arm B, girls-only HPV vaccination; or Arm C, no HPV vaccination. This is a massive effort in a well followed nordic population. 

The % of patients vaccinated within each cohort is hard to decipher as you read the manuscript. Please make this more clear in the text.

No estimate of the stability of the populations (i.e. how much intermingling) was provided, and should be.

In Figure 4, the drop in HSV2 seropositivity in Cohort B was as profound and significant as for the drop in HPV16/18 seropositivity in Cohort A. Please explain as this seems to call into question the main findings of the paper. 

They show that the HPV16 and HPV18 seroprevalence was decreased in young unvaccinated women after gender neutral vaccination, but with single gender vaccination under these conditions of low vaccination rates. This provides a demonstration of herd immunity, but is perhaps not all that surprising given that twice as many people in the gender neutral vaccination cohort were vaccinated. Nevertheless it shows the important message that when only low vaccination rates can be achieved in girls that there is value in vaccinating boys to improve protection of girls through herd immunity. It is also highly likely that boys will also receive direct health benefits themselves through prevention of anogenital cancers and head and neck cancers, and should be strongly urged anyway for reasons of equity.

Do you expect the impact of herd immunity to grow as the population ages?

Is it reasonable to consider looking at the impact on CIN2/3 rates in the medical record as well as seropositivity?

[LINK]

---

## [Decision Letter · Decision Letter 2]

6 Oct 2020

Dear Dr. Gray,

Thank you very much for submitting your revised manuscript "Gender-neutral vaccination is the key to HPV16 herd effect when vaccination coverage is limited: a cluster-randomized trial of vaccination strategy" (PMEDICINE-D-20-00112R2) for consideration at PLOS Medicine. 

Your revision was evaluated by a senior editor and discussed among all the editors here. It was also discussed with the academic editor, and sent to two of the original reviewersr. The reviews are appended at the bottom of this email and any accompanying reviewer attachments can be seen via the link below:

[LINK]

I am afraid that we still will not be able to accept the manuscript for publication in the journal in its current form, but we would like to consider a further revised version that addresses the reviewers' and editors' comments. Obviously we cannot make any decision about publication until we have seen the revised manuscript and your response, and we plan to seek re-review by one or more of the reviewers. 

We expect to receive your revised manuscript by Oct 27 2020 11:59PM. Please email us (plosmedicine@plos.org) if you have any questions or concerns.

We look forward to receiving your revised manuscript. 

Sincerely,

Thomas McBride, PhD

Senior Editor 

PLOS Medicine

plosmedicine.org

Comments from the Academic Editor:

The study is based on excellent data sources and a well-known and important HPV vaccine trial, with record linkage to a national cohort of and serum databank for pregnant women. Herd effects of HPV vaccination are a valid and useful outcome to examine through further analyses of the parent trial. But the revised version of this manuscript does not give enough information to understand motivation, planning, analysis plan or results adequately. After reading detailed and thoughtful comments of the reviewers, the actual changes to the manuscript are very limited. As one reviewer says, you need to go back to all the previous studies to piece things together. 

The paper is hard to follow for three main reasons. First, this report doesn’t explain well enough how this particular analysis fits in with the original trial, or the subsequently described plan to examine type replacement. Second, there are inconsistencies between this paper and the analysis plan that was provided. Third, it is not particularly easy to see how herd immunity is being defined and measured. I have a number of requests for clarification and suggestions.

General comments

1. Explanations and context: In the introduction, please explain the rationale for this study more fully, as it relates to the original objectives of the trial. The clinicaltrials.gov entry doesn’t mention type replacement or herd effects as primary or secondary outcomes, but the study design was reported as an RCT. The original RCT (as presented in ref 18) included modelling and provision to examine herd effects within the trial population, but did not specify additional analyses. Explain more clearly what the previous publications (esp refs 8 and 9) did and did not address about herd effects (e.g. lack of herd effects shown for HPV16 using HPV DNA positivity).

2. Without a protocol or statistical analysis plan written in advance for this particular analysis, it is hard to see whether the findings here showing support for a pre-defined hypothesis, or are exploratory and hypothesis-generating. In the introduction, please explain how the plan to assess herd effects of gender-neutral vaccination developed. The data from the Finnish Maternity Cohort (FMC) are mentioned in ref 18 for planning the trial, but not that they will be used in future. Explain how this study fits into the study analysis plan provided – that plan does not mention anything about measuring herd effects. The authors say they needed to find a herd effect before investigating type replacement. In this case, we would expect to see an analysis plan for investigating the herd effects before the analysis plan for type replacement. There is a new publication, which was not mentioned in the previous revisions, Gray P, et al. Int J Cancer 2020. Is this the manuscript that used the analysis plan? Please clarify, in the manuscript.

3. Given these limitations, I think that the title and conclusions (as stated in the Abstract and end of the main text) are somewhat overstated. The second bullet point of ‘What do these Findings Mean? in the Author summary is more appropriate as a conclusion.

4. Methods, study design could be clearer. The first two pages are confusing because they jump between the parent trial and the Finnish Maternity Cohort. 

4a. Please state the hypothesis, if there was one, or if not that these are exploratory analyses.

4b. Please state the primary outcome. I couldn’t find definitions of which was the primary and which were secondary outcomes. Similarly, results were not reported according to primary or secondary outcomes.

4c. Fig 1, please add a first level to the flow chart to show how the cohort studied in this trial is related to the parent trial, including total numbers in the communities in each arm. 

4d. Statistical analyses. The study is an analysis of a cohort nested in an RCT. Please describe how you measure follow-up and account for losses to follow-up. Please compare the characteristics of cohort participants to those of the female population of the same age in the trial communities. 

4e. The analysis plan provided says that the pre-vaccination period is 2005-2007 and the post-vaccination period is 2008-2016. In the analyses here, the pre-vaccination period is 2005-2010. But vaccination was given in 2007-2010, so these years are part of the vaccination and post-vaccination period. Please explain.

5. Herd effects. These can be hard to conceptualise and understand. 

5a. The authors now give a definition of herd protection. Could they give the full definition in the text that they gave in the response to reviewers? Could they also address Reviewer #1’s question about the relevant size of the difference? What size of difference was hypothesised?

5b. What was the vaccine coverage in the three arms? Is it possible that there is a greater effect in Arm A because overall coverage is higher? In Arm A, girls+boys coverage = 47.1% of girls + 19.5 of boys%. In Arm B, it’s 45.8% of girls.

5c. The authors report three different analyses, but these don’t seem to be comparable with respect to factors that are controlled for or stratified by. Please calrify these choices.

5c.i. Indirect effect of ‘community-wise’ vaccination over the study period – absolute seroprevalence in pre- and post-vaccination periods – why not controlled for smoking, or stratified by HSV-2 status? 

5c.ii. Further assess indirect effect of ‘community-wise’ vaccination in the post-vaccination era – within-arm seroprevalence ratios, comparing post-vaccination and pre-vaccination (controlled for smoking – but we should see the crude ratios too)

5c.iii. Overall indirect effect of gender-neutral and girls-only, compared with counterfactual – between-arm ratio of seroprevalence – ratio of ratios - ? controlled for smoking?

6. Discussion, Interpretation according to gender-neutral strategy: Is there any information on sexual mixing between girls and boys who are vaccinated or not vaccinated? How would this affect the findings if there were assortativity according to vaccination status?

Specific queries (page and line numbers as in the marked up copy because there are none in the clean version of the pdf)

1. ‘Community-wise’ – doesn’t make sense – should it be community-wide (across all communities)? Or community-specific (for a specific community)?

2. Arms A, B and C. It’s hard to remember which arm is which. In some parts of the manuscript, it would be easier to refer to the gender-neutral arm of the trial (maybe with Arm A in brackets).

3. P8, line 193: why ‘1982 and younger birth cohorts’? Should this be 1992?

4. P10, line 226: ‘due residing’ doesn’t make sense.

5. P10, line 231-2: ‘weighted to the birth cohort distribution found in each year…’ ‘distribution’ of what? Age? Sex?

6. P12, line 282-3: HSV-2 seroprevalence – please refer to Fig 4.

7. P13, lines 306-8: Figures in Table 1a and text don’t match.

8. Fig 3. Why is there no shading in the years 2007-2010, when HPV vaccination was offered as part of the trial? 

9. P15, line 346, ‘randomized trial evidence’ – please qualify this, given that the study population is a subset of the women in the communities.

10. P15: please explain why Chlamydia trachomatis as a marker of high sexual partner change (core group membership), would not be associated with a herd effect against HPV16 but HSV2 seroprevalence would? 

1- Thank you for updating the study design to cluster-randomized. However, following our email correspondence, it is now clear that this was a cohort study based on data from a cluster-randomized trial. Please update the Title and all other mentions of the study design throughout the manuscript to reflect this. 

2- Please also include the prospective analysis plan (from our email correspondence) as a supplemental file, referenced from the Methods section. Any changes in the analysis-- including those made in response to peer review comments-- should be identified as such in the Methods section of the paper, with rationale.

3- Thank you for including the Cluster Trials CONSORT extension. However, as an observational study, it is more appropriate to report according to the STROBE Guideline, and include the completed STROBE checklist as Supporting Information. Please add the following statement, or similar, to the first section Methods: "This study is reported as per the Strengthening the Reporting of Observational Studies in Epidemiology (STROBE) guideline (S1 Checklist)." and remove the CONSORT checklist (and its mention in the Methods)

4- Please describe briefly the ethical, legal, or contractual restriction that prevents you from sharing the deidentified dataset publicly. Please also note that a study author cannot be the contact person for the data. Please provide a different contact for data access, such as a member of your institute’s data or ethics committee. 

5-Please revise your title according to PLOS Medicine's style. Your title must be nondeclarative and not a question. I recommend "Human papillomavirus seroprevalence in pregnant women following gender neutral and girls-only vaccination programs in Finland: a cohort analysis following a cluster randomized trial.”

6- Throughout the manuscript, please replace “subjects” with “participants”.

7- Abstract Background: Please be more specific about what you mean by “medically most important HPV type”, and cite the actual reproductive number instead of “high”.

8- In addition to stating the number of communities in each arm, please also include the number of participants in each arm when first describing the study in the Abstract.

9- In the Abstract and throughout the Results, please report all main outcomes (i.e., seroprevalence differences for all HPV types).

10- Thank you for mentioning a study limitation at the end of the Abstract Methods and Findings. Please make this a separate sentence(s), and include other relevant study limitations (e.g., generalizability outside Finland, potential effects of secular trends in sexual behaviors).

11- Abstract Conclusions: Please address the study implications without overreaching what can be concluded from the data; the phrase "In this study, we observed ..." may be useful.

12- Additionally, in the Abstract Conclusions, please interpret the study based on the results presented in the abstract, emphasizing what is new without overstating your conclusions. Perhaps, “These findings suggest that gender-neutral vaccination can facilitate the control of HPV16…”

13- Author summary point 3, please edit to: “Achieving herd protection for HPV16 requires greater than 80% girls-only vaccination coverage, a level that has not been achievable in many countries.”

14- Author Summary point 4 (“Evidence-based solutions to this problem are required...”) should be removed and replaced with the study question.

15- Author Summary point 7 (“Only among the unvaccinated females...”) is phrased a bit awkwardly, consider editing.

16- The last sentence of the Introduction could be split into two sentences to improve clarity.

17- For all observational studies, in the manuscript text, please indicate: (1) the specific hypotheses you intended to test, (2) the analytical methods by which you planned to test them, (3) the analyses you actually performed, and (4) when reported analyses differ from those that were planned, transparent explanations for differences that affect the reliability of the study's results. If a reported analysis was performed based on an interesting but unanticipated pattern in the data, please be clear that the analysis was data-driven.

18- Please remove trademark symbols (e.g., CervarixTM, Engerix®-B).

19- Thank you for including vaccination coverage. Please include the numerators and denominators that go along with each percentage. Additionally, this seems like it should be in the Results rather than Methods?

20- Please provide a table showing the baseline characteristics of the study population.

21- Regarding the point (made by reviewers 1 and 4) about the likelihood of intermingling among the study population, your response should be noted in the manuscript text, in case readers also have the same question. Also, is there a reference you could point to on this point?

22- Figure 5, the color legend is over top of the bottom ratio numbers.

23- Please integrate your response to Reviewer 4’s 4th point (about the HSV seroprevalence) into the main text, as readers may have the same question.

24- Please include a table of absolute numbers of seropositive cases, in addition to the information on adjusted seroprevalence ratio contained in the current table 1.

25- Please incorporate table S3 into the main paper.

26- Please begin the Discussion with a brief summary of what was done, before describing the results.

27- The phrase “sexual risk-taking core group” still appears, please replace with a less stigmatizing phrase.

28- Please rephrase the final paragraph of the Discussion to limit the conclusions to the current study. E.g., “Our results suggest that when HPV vaccination coverage is moderate…”

Comments from the reviewers:

Reviewer #1: We thank the authors for considering the points raised in our previous review. They have largely been addressed, and only a few suggestions remain:

1. The additional advantages of the new cumulative incidence methodology over the previous one-time PCR-positivity might be briefly mentioned at the end of the Introduction section, for readers to more quickly recognize the significance of the work. The relevant previous work might also be cited there.

2. The (low) possibility of movement between communities might also be briefly mentioned as a possible limitation.

3. The addition of Table S1 is much appreciated. If possible, it might be briefly explained as to how the sensitivity/specificity used in the analysis, were derived from the primary assumption.

Reviewer #3: Great work by the authors. Remaining comments:

* It is recommended in flow charts by CONSORT to include all those assessed for eligibility in the study and document reasons for exclusion from the study. Please include in the flowchart all women <23 in the Finnish Maternity Cohort in the communities, and indicate how many were excluded due to lack of consent and the 3498 excluded due to registry-linked HPV vaccination status by study period in Figure 2.

* Please indicate what cutoff was used when assessing HPV vaccination status through antibody titers.

* They repeat the definition of exposure twice in the methods, this sentence could be deleted: "Exposure to the indirect effects of HPV16/18 vaccination is defined as residing at the time of sample donation in one of the community-randomized HPV vaccination trial communities."

[LINK]

---

## [Decision Letter · Decision Letter 3]

15 Mar 2021

Dear Dr Gray, 

On behalf of my colleagues and the Academic Editor, Nicola Low, I am pleased to inform you that we have agreed to publish your manuscript "Human papillomavirus seroprevalence in pregnant women following gender-neutral and girls-only vaccination programs in Finland: a cross-sectional cohort analysis following a cluster-randomized trial" (PMEDICINE-D-20-00112R3) in PLOS Medicine.

PRESS

OPEN SCIENCE 

Sincerely, 

Dr Raffaella Bosurgi 

Executive Editor 

PLOS Medicine